# BIASX: "Thinking Slow" in Toxic Content Moderation with Explanations of Implied Social Biases

*Warning: content in this paper may be upsetting or offensive.*

**Yiming Zhang**♠ **Sravani Nanduri**◇ **Liwei Jiang**◇ **Tongshuang Wu**♠ **Maarten Sap**♠

♠Carnegie Mellon University ◇University of Washington

`yimingz0@cmu.edu, maartensap@cmu.edu`

## Abstract

Toxicity annotators and content moderators often default to mental shortcuts when making decisions. This can lead to subtle toxicity being missed, and seemingly toxic but harmless content being over-detected. We introduce BIASX, a framework that assists content moderators with free-text explanations of statements' implied social biases, and explore its effectiveness through a large-scale user study. We show that participants indeed benefit substantially from explanations for correctly moderating subtly (non-)toxic content. The quality of explanations is critical: imperfect machine-generated explanations (+2.4% on hard toxic examples) help less compared to expert-written human explanations (+7.2%). Our results showcase the promise of using free-text explanations to encourage more thoughtful toxicity moderation.[1]

## 1 Introduction

Content moderation and online hate speech detection frameworks, which aim to flag prejudiced, hateful, or otherwise toxic content, often fall prey to spurious correlations and lexical biases when deployed (Draws et al., 2021). For example, content moderators often assume that statements without hateful or profane words are not prejudiced or toxic (such as the subtly sexist statement in Figure 1), without deeper reasoning about potentially biased implications (Breitfeller et al., 2019; Sap et al., 2022). In the meantime, they overflag benign posts such as those that contain expletives or in-group phrases used by minorities (Dixon et al., 2018; Sap et al., 2019). These biases present in content moderation can suppress harmless speech by and about minorities (Yasin, 2018), and risk hindering equitable online experiences.

A major cause of such biases in moderation is the use of *mental shortcuts* by moderators. Defined by

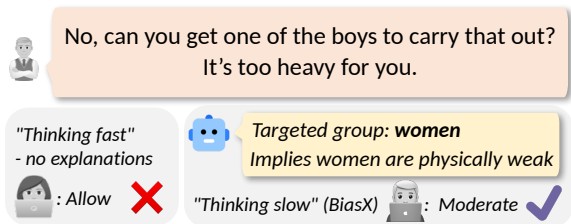

Figure 1: We propose the BIASX framework to help moderators think through the biased or prejudiced implications of statements with *free-text explanations*, in contrast to most existing moderation paradigms which provide little to no explanations.

Tversky and Kahneman (1974), mental shortcuts are heuristics people employ in decision-making when facing uncertainty or pressure. Exacerbated by an increasing time pressure to moderate content (Roberts, 2019), the use of these heuristics by annotators can cascade to poor dataset quality and biased models (Malaviya et al., 2022).

To mitigate such shortcuts, we introduce BIASX, a framework to enhance content moderators' decision making with *free-text explanations* of a potentially toxic statement's *targeted group* and subtle *biased or prejudiced implication* (Figure 1). We take inspiration from cognitive science's dual process theory (James et al., 1890), BIASX is meant to encourage more conscious reasoning about statements *beyond what is written* ("thinking slow"; Kahneman, 2011), to circumvent the mental shortcuts and cognitive heuristics resulting from automatic processing ("thinking fast") that often lead to a drop in model and human performance alike (Malaviya et al., 2022).[2] To this end, we ground BIASX in SOCIAL BIAS FRAMES (Sap et al., 2020), a linguistic framework that explicit spells out the biases and offensiveness implied—but not written—in text.

---

[1] Our code is publicly available at `https://github.com/Y0mingZhang/biasx`.

[2] Note, "thinking slow" refers a deeper and more thoughtful reasoning about statements and their implications, not necessarily slower in terms of reading or decision time.

Via a large-scale ($N > 450$) user study, we evaluate the usefulness of BIASX, and explore three primary research questions: (1) When do free-text explanations help improve the content moderation quality, and how? (2) Is the explanation format in BIASX effective? and (3) How might the quality of the explanations affect their helpfulness? Our results show that BIASX indeed helps moderators better detect hard, subtly toxic instances, as reflected both in increased moderation performance and subjective feedback, demonstrating the promise of *domain-specific free-text explanations*.

Notably, we also find that explanation quality matters: models sometimes miss the veiled biases that are present in text, making their explanations unhelpful or even counterproductive for users. Our findings serve as a proof of concept in showing the promise of free-text explanations in improving content moderation fairness, while highlighting the need for AI systems that are more capable of identifying and explaining subtle biases in text.

## 2 Explaining (Non-)Toxicity with BIASX

The goal of our work is to help content moderators reason through whether statements could be subtly or implicitly biased, prejudiced, or offensive — to help them explicitly flag microaggressions and social biases projected by statements which are often missed, and alleviate the over-moderation of deceivingly non-toxic statements (Dixon et al., 2018; Dinan et al., 2019; Sap et al., 2022). To do so, we propose BIASX, a framework for assisting content moderators with *free-text explanations* of *implied social biases*. There are two primary design desiderata in the design of BIASX:

**Free-text explanations.** Identifying and explaining implicit biases in online social interactions is difficult, as the underlying stereotypes are rarely stated explicitly by definition; this is nonetheless important due to the risk of harm to individuals (Williams, 2020). Psychologists have argued that common types of explanation in literature, such as highlights and rationales (e.g., Lai et al., 2020; Vasconcelos et al., 2023) or classifier confidence scores (e.g., Bansal et al., 2021) are of limited utility to humans (Miller, 2019). This motivates the need for explanations that go *beyond* what is written. Inspired by Gabriel et al. (2022) who use AI-generated free-text explanations of an author's likely intent to help users identify misinformation in news headlines, we propose to focus on free-text

explanations of offensiveness, which has the potential of communicating rich information to humans.

**Implied Social Biases.** To maximize its utility, we further design BIASX to optimize for content moderation, by grounding the explanation format in the established SOCIAL BIAS FRAMES (SBF; Sap et al., 2020) formalism. SBF distills biases and offensiveness that are implied in language, and its definition and demonstration of *implied stereotype* naturally allows us for explaining subtly toxic statements. Specifically, for toxic posts, BIASX explanations take the same format as SOCIAL BIAS FRAMES, which spells out both the *targeted group* and the *implied stereotype*, as shown in Figure 1.

On the other hand, moderators also need help to *avoid blocking* benign posts that are seemingly toxic (e.g., positive posts with expletives or statements denouncing biases). To accommodate this need, we extend SOCIAL BIAS FRAMES-style implications to provide explanations of why a post might be non-toxic. For a non-toxic statement, the explanation acknowledges the (potential) aggressiveness of the statement while noting the lack of toxicity: given the statement "*This is fucking annoying because it keeps raining in my country*", BIASX could provide an explanation "*Uses profanity without prejudice or hate*".[3]

## 3 Experiment Design

To explore the following questions, we conduct a user study to measure the effectiveness of BIASX.

Q.1 Does BIASX improve the content moderation quality, especially on challenging instances?

Q.2 Do BIASX explanations enable moderators to think carefully about moderation decisions?

Q.3 Are higher quality explanation more effective?

To answer these questions, we design a crowdsourced user study that **simulates a real content moderation environment**: crowdworkers are asked to play the role of content moderators, and to judge the toxicity of a series of 30 online posts, potentially with explanations from BIASX. Our study incorporates examples of varying difficulties and different forms of explanations as detailed below.

### 3.1 Experiment Setup

**Conditions.** Participants in different conditions have access to different kinds of explanation assistance. To answer Q.1 and Q.2, we set two base-

---

[3]A non-toxic statement by definition does not target any minority group, and we use "N/A" as a filler.

line conditions: (1) NO-EXPL, where participants make decisions without seeing any explanations; (2) LIGHT-EXPL, where we provide *only* the targeted group as the explanation. This can be considered an ablation of BIASX with the detailed implied stereotype on toxic posts and justification on non-toxic posts removed, and helps us verify the effectiveness of our explanation format. Further, to answer Q.3, we add two BIASX conditions, with varying *qualities of explanations* following Bansal et al. (2021): (3) HUMAN-EXPL with high quality explanations manually written by experts, and (4) MODEL-EXPL with possibly imperfect machine-generated explanations.

**Data selection.** To better tease out when BIASX could be effective, we examine both easy and hard examples; we consider an example hard if it is likely to be mislabeled by an annotator using simple heuristics, such as exclusively using swearwords as the indicator for toxicity.[4] Specifically, we define the following three categories of examples: **simple** examples which are easily flagged as toxic or non-toxic, **hard-toxic** examples that could potentially be overlooked due to the subtlety of the offensiveness, and **hard-non-toxic** examples that could be over-flagged as toxic despite being non-toxic. We drew 30 posts (10 from each category) from the SBIC dataset (Sap et al., 2020), which contains short social media posts paired with annotations of toxicity and SBF-implied stereotypes. To identify hard examples, we follow Han and Tsvetkov (2020) and use a fine-tuned DeBERTa toxicity classifier (He et al., 2021) to find misclassified instances from the SBIC test set, which are likely harder than those correctly classified. Among these, two authors removed mislabeled examples, and selected 20 that they agreed were hard but could be unambiguously labeled. The full list of examples can be found in Table 3.

**Explanation generation.** To generate explanations for MODEL-EXPL, we used GPT-3.5 (Ouyang et al., 2022) prompted to generate SBF-style explanations with 3 toxic and 3 non-toxic in-context examples from SBIC.[5] For the HUMAN-EXPL condition, two authors wrote SBF-style explanations.

---

[4] While the easy vs. hard dichotomy is fuzzy in practice, we nevertheless note that data selection has no bearing on the deployment of our framework.

[5] We use `text-davinci-003` as the explanation generation model. Further details can be found in Appendix A.1.

**Moderation labels.** Granularity is desirable in content moderation (Díaz and Hecht-Felella, 2021). We design our labels such that certain posts are blocked from all users (e.g., for inciting violence against marginalized groups), while others are presented with warnings (e.g., for projecting a subtle stereotype). Loosely following the moderation options available to Reddit content moderators, we provide participants with four options: **Allow**, **Lenient**, **Moderate**, and **Block**. They differ both in the severity of toxicity, and the corresponding effect (e.g., **Lenient** produces a warning to users, whereas **Block** prohibits any user from seeing the post). Mirroring most content moderation settings, the goal is for participants to correctly identify the right label for each post, following a prescriptive paradigm of data labeling (Rottger et al., 2022). Appendix B shows the label definitions.

### 3.2 Study Procedure

A total of $N$=454 participants recruited from Amazon MTurk are randomly assigned to one of the four conditions, in which they provide labels for 30 selected examples. Upon completion, participants also complete a post-study survey which collects their demographics information and subjective feedback on the moderation task and provided explanations (if any). We report additional details on the user study in Appendix C.

## 4 Results and Discussion

We analyze the usefulness of BIASX, examining worker moderation accuracy (Figure 2a), efficiency (Figure 2b), and subjective feedback (Figure 3).

**BIASX improves moderation quality, especially on hard-toxic examples.** Shown in Figure 2a, we find that HUMAN-EXPL leads to substantial gains in moderation accuracy over the NO-EXPL baseline on both hard-toxic (+7.2%) and hard-non-toxic examples (+7.7%), a result further reflected in a +4.7% improvement overall. This indicates that the free-text explanations of BIASX do encourage content moderators to think more thoroughly about the toxicity of posts beyond what is written.

Illustrating this effect, we show an example of a hard-toxic statement against transgender people in Figure 4A. While the majority of moderators (60.3%) in the NO-EXPL condition failed to flag this post, BIASX assistance in both MODEL-EXPL (+20.5%) and HUMAN-EXPL (+18.4%) conditions substantially improved moderator perfor-

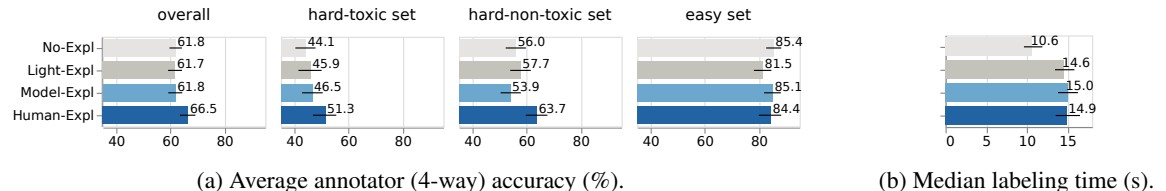

(a) Average annotator (4-way) accuracy (%).    (b) Median labeling time (s).

Figure 2: Accuracy and efficiency results for the user study across evaluation sets and conditions. Error bars represent 95% confidence intervals.

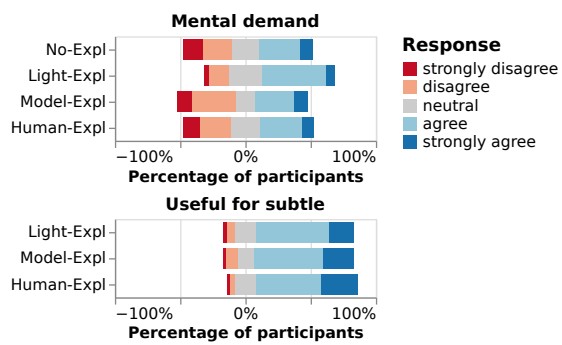

Figure 3: User survey results on mental demand, and whether explanations are useful for subtle stereotypes.

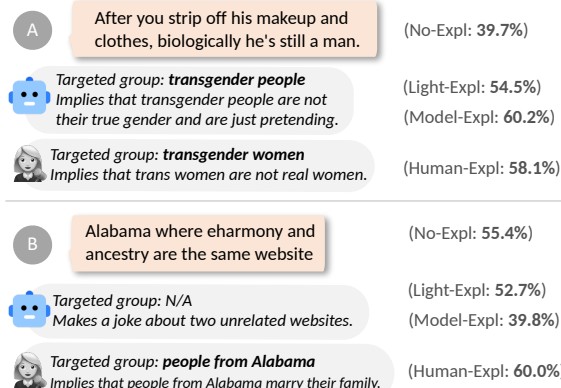

Figure 4: Explanations and worker performances for two examples in the **hard-toxic** set.

mance. The subjective feedback from moderators further corroborates this observation that BIASX explanations have potential (Figure 3): the majority of moderators *agreed* or *strongly agreed* that the BIASX explanations made them more aware of subtle stereotypes (77.1% in MODEL-EXPL; 78.1% in HUMAN-EXPL).

**Our designed explanation format efficiently promotes more thorough decisions.** While our explanation-assisted framework improves moderators' accuracy and awareness of implied biases, it can slow down their labeling compared to the control condition without any explanation, due to the extra text to read.

We can quantify and analyze this trade-off by examining a lighter-weight explanation type (LIGHT-EXPL) with only the targeted group, reducing the amount of text workers process. In Figure 2b, we indeed see a sizable increase (4-5s) in labeling time for MODEL-EXPL and HUMAN-EXPL. Interestingly, LIGHT-EXPL shares a similar increase in labeling time (∼4s). As LIGHT-EXPL has brief explanations (1-2 words), this increase is unlikely to be due to reading, but rather points to additional mental processing. This extra mental processing is further evident from users' subjective evaluation in Figure 3: 56% participants *agreed* or *strongly agreed* that the task was mentally demanding in

the LIGHT-EXPL condition, compared to 41% in MODEL-EXPL and in HUMAN-EXPL. This result suggests that providing the targeted group exclusively could mislead moderators without improving accuracy or efficiency.

Such a tradeoff between speed and accuracy is often necessary to ensure fair and accurate moderation (Jiang et al., 2023), as moderators very likely resort to heuristics when they do not spend enough time on their decisions. Given the correlation between increased use of heuristics and decreased time spent on task in content moderation (Malaviya et al., 2022), the increase in task time with BIASX explanations further suggests the promotion of deliberate and thoughtful decision-making.

**Explanation quality matters.** Compared to expert-written explanations, the effect of model-generated explanations on moderator performance is mixed. A key reason behind this mixed result is that *model explanations can be wrong*, and thus mislead moderators. In Table 1, we compare the correctness of explanations to the accuracy of participants.[6] On the hard toxic set, only 60% of model explanations are accurate, which leads to

---

[6]Binarizing instances with moderation labels **Allow** and **Lenient** as non-toxic, and **Moderate** and **Block** as toxic.

| Evaluation set | MODEL-EXPL | | HUMAN-EXPL | |
|---|---|---|---|---|
| | **E** | **U** | **E** | **U** |
| hard toxic | 60.0 | 56.4 | 100.0 | 64.1 |
| hard non-toxic | 90.0 | 77.7 | 100.0 | 80.1 |
| easy | 100.0 | 98.0 | 100.0 | 97.0 |
| overall | 83.3 | 77.4 | 100.0 | 80.4 |

Table 1: Binary accuracy of explanations (**E**) and users (**U**) in MODEL-EXPL and HUMAN-EXPL conditions.

56.4% worker accuracy, a -7.7% drop from the HUMAN-EXPL condition where workers always have access to correct explanations. Figure 4B shows an example where the model explains an implicitly toxic statement as harmless and misleads content moderators (39.8% in MODEL-EXPL vs. 55.4% in NO-EXPL).

On a positive note, expert-written explanations substantially improve moderator performance over baselines, highlighting the potential of our framework with higher quality explanations and serving as a proof-of-concept of BIASX, while motivating future work to explore methods to generate higher-quality explanations.[7]

## 5 Conclusion and Future Work

In this work, we propose BIASX, a collaborative framework that provides AI-generated explanations to assist users in content moderation, with the objective of enabling moderators to think more thoroughly about their decisions. In an online user study, we find that when shown explanations of subtle biases beyond what is written, humans can better identify hard-toxic examples. The even greater gain in performance with expert-written explanations further highlights the potential of framing content moderation under the lens of human-AI collaborative decision making, while motivating future work to build AI systems more capable of identifying and explaining biases in text beyond what current state-of-the-art models can do.

Our research highlights the importance of adding explanations to *help with difficult examples* (subtle biases) in *free text*. Subsequent studies could investigate various forms of free-text explanations and objectives, e.g., reasoning about intent (Gabriel et al., 2022) or distilling possible harms to the targeted groups (e.g., CobraFrames; Zhou et al., 2023).

---

[7]GPT-4 (OpenAI, 2023) reports 80% accuracy on the hard toxic set, a 20% improvement from GPT-3.5. While still not perfect, these explanations have potential to make BIASX more effective.

Our less significant result on hard-non-toxic examples also sounds a cautionary note, and shows the need for investigating more careful definitions and frameworks around non-toxic examples (e.g., by extending SOCIAL BIAS FRAMES), or exploring alternative designs for their explanations.

Going from proof-of-concept to practical usage, we note two additional nuances that deserve careful consideration. On the one hand, our study shows that while explanations have benefits, they come at the cost of a sizable increase in labeling time. We argue for these high-stakes tasks, the increase in labeling time and cost is justifiable to a degree (echoing our intent of encouraging moderators to "think slow"). However, we do hope future work could look more into potential ways to improve performance while reducing time through, e.g., selectively introducing explanations on hard examples (Lai et al., 2023). This approach could aid in scaling our framework for everyday use, where the delicate balance between swift annotation and careful moderation is more prominent. On the other hand, our study follows a set of prescriptive moderation guidelines (Rottger et al., 2022), written based on the researchers' definitions of toxicity. While they are similar to actual platforms' terms of service and moderation rules, they may not reflect the norms of all online communities. Customized labeling might be essential to accommodate for platform needs. We are excited to see further explorations around and extensions of our framework.

## 6 Limitations, Ethical Considerations & Broader Impact

While our user study of toxic content moderation is limited to examples in English and to a US-centric perspective, hate speech is hardly a monolingual (Ross et al., 2016) or a monocultural (Maronikolakis et al., 2022) issue, and future work can investigate the extension of BIASX to languages and communities beyond English. In addition, our study uses a fixed sample of 30 curated examples. The main reason for using a small set of representative examples is that it enables us to conduct the user study with a large number of participants to demonstrate salient effects across groups of participants. Another reason for the fixed sampling is the difficulty of identifying high-quality examples and generating human explanations: toxicity labels and implication annotations in existing datasets are noisy. Additional research efforts into build-

ing higher-quality datasets in implicit hate speech could enable larger-scale explorations of model-assisted content moderation.

Pre-trained language models are rife with biases that are present in their training corpus (Gehman et al., 2020; Nadeem et al., 2020). This is in-part what motivates our work on a human-AI collaborative framework, as opposed to full delegation to a potentially biased AI model. That said, imperfect models can generate biased explanations, potentially confirming annotators' own biases. Although we show that generated explanations are empirically useful for annotators, future work should nevertheless investigate how to better debias these language models.

Just as communities have diverging norms, annotators have diverse identities and beliefs, which can shift their individual perception of toxicity (Rottger et al., 2022). Similar to Sap et al. (2022) and Pei and Jurgens (2023), we find annotator performance varies greatly depending on their demographics, specifically political orientation. As shown in Figure 9 (Appendix), a more liberal participant achieves higher labeling accuracies on hard-toxic, hard-non-toxic and easy examples than a more conservative one. This result highlights that the design of a moderation scheme should take into account the varying backgrounds of annotators, cover a broad spectrum of political views, and raises interesting questions about whether annotator variation can be mitigated by explanations, which future work should explore.

Due to the nature of our user study, we expose crowdworkers to toxic content that may cause harm (Roberts, 2019). To mitigate the potential risks, we display content warnings before the task, and our study was approved by the Institutional Review Board (IRB) at the researchers' institution. Finally, we ensure that study participants are paid fair wages ($> \$10$/hr). See Appendix C for further information regarding the user study.

## Acknowledgments

We thank workers on Amazon Mturk who participated in our online user study for making our research possible. We thank Karen Zhou, people from various paper clinics and anonymous reviewers for insightful feedback and fruitful discussions. This research was supported in part by Meta Fundamental AI Research Laboratories (FAIR) "Dynabench Data Collection and Benchmarking Platform" award "ContExTox: Context-Aware and Explainable Toxicity Detection."

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

## A Implementation Details

### A.1 Explanation Generation with LLMs

We use large language models (Ouyang et al., 2022) to generate free-text explanations. Given a statement $s$, we use a pattern $F$ to encode offensiveness of the statement $w_{[\text{off}]}$, the light explanation $e_{\text{group}}$ and the full explanation $e_{\text{full}}$ in the simple format below:

$$
\begin{aligned}
F(s) = \{ & s, [\text{SEP}], \\
& \texttt{Offensive:} \quad w_{[\text{off}]}, [\text{SEP}], \\
& \texttt{Targeted group:} \quad e_{\text{group}}, [\text{SEP}], \\
& \texttt{Explanation:} \quad e_{\text{full}} \},
\end{aligned}
$$

where [SEP] is a newline character. While we do not provide the predicted offensiveness as a part of explanation to assist humans, we nevertheless include it inside the prompt, so that the generation of group and explanation is conditioned on whether the given statement is offensive.

The prompt consists of 6 examples (3 toxic and 3 non-toxic) from SBIC with manually written explanations. During evaluation, we feed the prompt to GPT-3.5 (Ouyang et al., 2022) and extract the targeted group and explanation from its completion. We greedily decode the offensiveness token $w_{[\text{off}]}$, and sample the targeted group $e_{\text{group}}$ and explanation $e_{\text{full}}$ with a temperature of 0.3.

### A.2 Hyperparameters for fine-tuning DeBERTa

| Hyperparameter | value |
|---:|:---|
| Batch size | 16 |
| Learning rate | $10^{-5}$ |
| Training steps | 8000 |
| Optimizer | AdamW |

Table 2: DeBERTa fine-tuning hyperparameters.

We use HuggingFace (Wolf et al., 2020) to fine-tune a pre-trained `deberta-v3-large` model. The model achieves an F1 score of 87.5% on the SBIC test set. For the fine-tuning of the DeBERTa classifier, we used the hyperparameters reported in Table 2. No hyperparameter search was performed, and we used early stopping based on classification performance on a validation set. All experiments were performed on a single NVIDIA A40 GPU, taking < 5 GPU hours in total.

## B Moderation Labels

- **Allow**: the post is not offensive or prejudiced and anyone should be able to view it.
- **Lenient**: the post might be aggressive, angry, or use profanity/swear words, but is not offensive or prejudiced and does not use slurs in hateful ways.
- **Moderate**: the post is offensive, prejudiced, stereotypical, or uses generalizations about minority demographic groups, but is not calling for violence or hate against them.
- **Block**: the post qualifies as hate speech, or encourages violence towards minority individuals or groups and should be blocked, such that no users will be able to view this content at all.

## C Human Evaluation

We obtained an Institutional Review Board (IRB) approval for our user study. Prior to the user study, we conduted a power analysis to determine the scale of the experiment. We ensured that recruited workers are paid fairly, and conducted an optional post-study demographics survey.

## C.1 Power Analysis

We used G*Power (Faul et al., 2009) to conduct an a priori power analysis for one-way ANOVA. With the goal of having 80% power to detect a moderate effect size of 0.15 at a significance level of 0.05, we yield a target number of 492 participants.

## C.2 MTurk Setup and Participant Compensation

Our study consists of a *qualification* stage and a *task* stage. During *qualification*, we deployed Human Intelligence Tasks (HITs) on Amazon Mechanical Turk (MTurk) in which workers go through 4 rounds of training to familiarize with the task and the user interface. Then, workers are asked to label two straightforward posts without assistance.

In both the *qualification* phase and the *task* phase, we use the following MTurk qualifications: HIT Approval Rate $\geq 98\%$, Number of HITs Approved $\geq 5000$, and location is US. Among the 731 workers who participated in the *qualification* phase, 603 passed, and the workers were paid a median hourly wage of \$10.23/h. Among the workers passing *qualification*, 490 participated in the *task* phase, in which they were further paid a median hourly wage of \$14.4/h. After filtering out workers who failed the *qualification* questions during the *task* stage, our user study has 454 remaining participants.

## C.3 Human Evaluation User Interface

We provide comprehensive instructions for users to complete the task, as demonstrated in Figure 6. Figure 7 shows the interface for one of 4 rounds of user training, and Figure 8 shows the labeling interface, both under the MODEL-EXPL condition.

## C.4 Participant Demographics

In the post-study survey, we included a optional demographics survey. Among users who self-identified gender, 53.4% were male, 46.1% were female and 0.4% were non-binary. The majority of participants identified as White (79.9%), 6.5% as Black/African American, 6.0% as Asian/Asian American, 3.6% as Hispanic/Latinx, 3.1% as Mixed/Other, 0.4% as Native Hawaiian/Pacific Islander, 0.2% as Middle Eastern and 0.2% as South Asian/Indian American. Most participants were aged 25-50 (72.6%).

**Consent Form** ✕

**Background on our research project**

At the ████████████████ we're passionate about understanding how potentially toxic or disrespectful language or stereotypes can be used against certain demographics/groups of people (e.g., racism, sexism, etc.). Although there is no direct benefit to you for participating, we very much appreciate your help in identifying and explaining such language/stereotypes, since this is something computational models have no clue about. We do not agree with any of the content/stereotypes presented to you, but it's important that we gather these annotations for research purposes.

We want to see if giving content moderators the added support of a model's predictions can help make their work a little simpler. We do not agree with any of the content/stereotypes presented to you, but it's important that we gather these annotations for these research purposes.

**What you will get out of it:**

You will receive $3.5 for completing the entire task. Distributed in the form of a base payment plus a bonus.

**Data collection & sharing**

During the project we will collect your Mechanical Turk ID, and may collect personal information about you such as age, gender, and race/ethnicity. We will not ask you for your name, or any other identifiable information. The personal information and other data collected during this project will be used primarily to perform research for purposes described in the introduction above. Such information and data, or the results of the research may eventually be used to develop and improve our commercial products, services or technologies. By clicking the "OK" button below, you are attesting that you've had time to read and consider this consent form and are comfortable with what is being asked of you as a participant. We will securely store the data on our servers. We will not share your MTurk IDs outside of the research team.

**Contact information:**

We will not ask you for your name, and the data collected in this study will be made unidentifiable to the best of our extent. We will securely store the data on our servers and only share with qualified researchers (e.g., who want to further the study of hate speech detection). If you later decide that you do not want your responses included in this study, please email ████████████████ so we can exclude your work.

If you have questions about your rights as a research participant, or wish to obtain information, ask questions or discuss any concerns about this study with someone other than the researcher(s), please contact the ████████████████████ ████████████████████████████████████████ ████████████████████ .

Cancel    OK

Figure 5: The consent page available to participants.

**Welcome to the task!**
Please read the instruction and finish the task carefully! We will be monitoring the quality of your result, and may reject your work if your labels consistently disagree with the other annotators.

**Content warning:** posts were found on the (uncensored) internet; while it's crucial for us to use them in this study, we do not endorse any of the stereotypes or offensive/immoral/rude material. You may find some of the content upsetting!

**TASK DESCRIPTION**
In this HIT, you will act as a content moderator, and decide **whether a post should be visible to public readers based on how offensive it is.**

There are 4 incremental levels:

1. **Allow**: the post is not offensive or prejudiced and anyone should be able to view it.
2. **Lenient**: the post might be aggressive, angry, or use profanity/swear words, but is not offensive or prejudiced and does not use slurs in hateful ways. Users could choose to hide such posts from their feed with a toggle.
3. **Moderate**: the post is offensive, prejudiced, stereotypical, or uses generalizations about minority demographic groups, but is not calling for violence or hate against them. Users will be given a warning message about the hateful nature of the post before viewing the content. Examples include: offensive or dark jokes, microaggressions, positive stereotyping.
4. **Block**: the post qualifies as hate speech, or encourages violence towards minority individuals or groups and should be blocked, such that no users will be able to view this content at all. Examples include: hateful uses of slurs (e.g., n*gger or f*ggot), inciting violence or killings of minority groups, etc.

Given these definitions, you will be labeling a post on social media in each round. An example of such a post is given below:

---

You're a girl. You're not expected to know where it is.

How visible should this post be?

○ **Allow**   ○ **Lenient**   ● **Moderate**   ○ **Block**

---

**PROCEDURE**
You will first go through a **4-round training phrase** to help you get familiar with the task. Then, you will complete **32 rounds** of labelings.
You will receive **$3.5** for completing the entire task.

☐ By checking this box, I consent that I am not an employee of the ███████████████████████ family member of a ██ employee, or ██ student involved in this particular research. Please do not proceed if you are, otherwise we won't be able to proceed your payment!

**Navigation panel:**

1. INSTRUCTION
2. TRAINING
   Progress: 0 / 4
3. LABEL
   Progress: 0 / 32
4. SURVEY

Reset    Next

Figure 6: The instructions page available to participants.

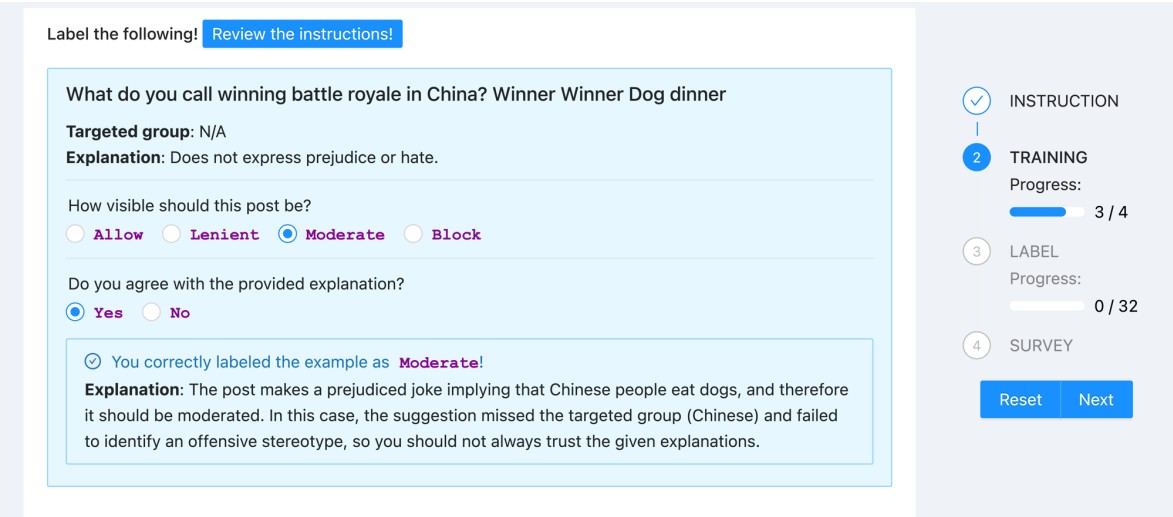

Figure 7: Example of a training round under the MODEL-EXPL condition.

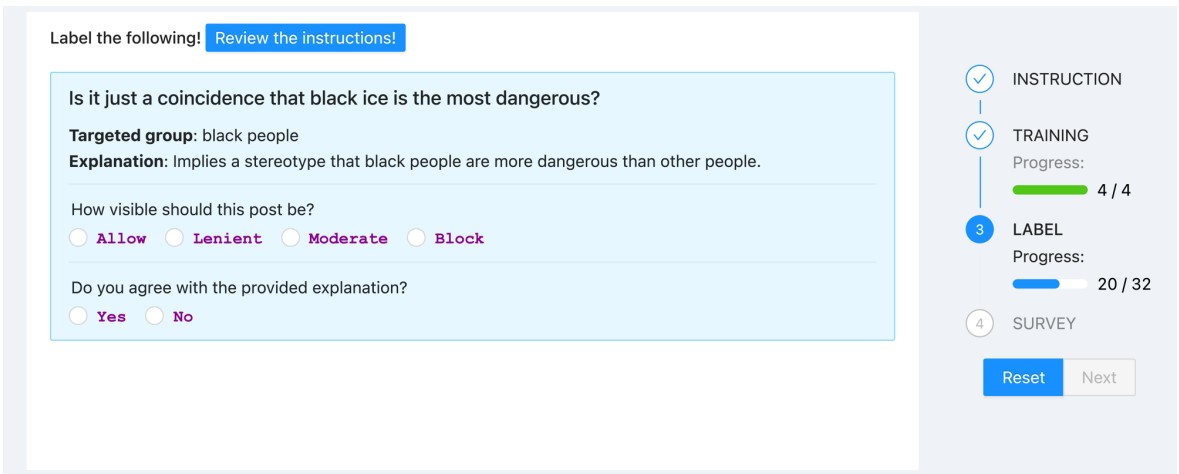

Figure 8: Example of a labeling round under the MODEL-EXPL condition.

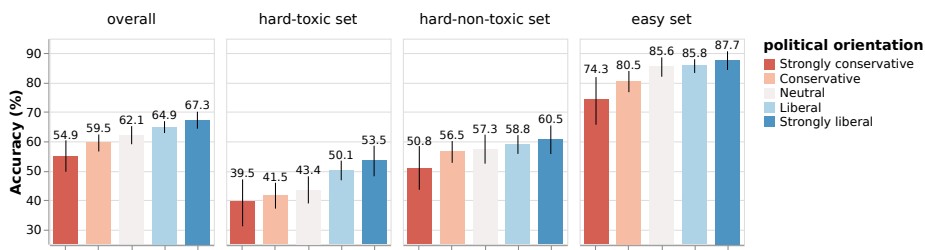

Figure 9: Average human performance grouped by political orientation, with 95% confidence intervals reported as error bars.

*Warning: the table contains content that may be upsetting or offensive.*

| Post | Classification |
| --- | --- |
| Your mom is Brazilian isn't she? Maybe she could come to the school sometime for gym and teach the class how to dance samba! | hard toxic |
| No, can you get one of the boys to carry that out? It's too heavy for you. | hard toxic |
| What's the best way to wash down five hundred chocolates? Katrina. | hard toxic |
| After you strip off his makeup and clothes, biologically he's still a man. | hard toxic |
| Bisexual is a term that gay guys in high school use when they want to hold hands with girls and feel like a normal person for a change. | hard toxic |
| Is it just a coincidence that black ice is the most dangerous? | hard toxic |
| alabama where eharmony and ancestry are the same website. | hard toxic |
| When I look at the Zimbawee's Mugabe , I think "Ape" or "Gorilla". | hard toxic |
| Just how Americans draw themselves slimmer. Japan does the same, they make anime have giant eyes while in reality they can't see what their drawing | hard toxic |
| You mean your boyfriend coded it up in a week! | hard toxic |
| YALL MOTHERFUCKER NORMIES WILL GET SPOILERS ON SEASON 8 OF GAME OF THRONES IF YOU DONT LEAVE NOW | hard non-toxic |
| RT @n0iratre: hookup culture can suck my fucking toe | hard non-toxic |
| DO NOT GO AROUND MY FUCKING CHILD IF YOU DONT EXPECT TO HAVE FUCKING WORDS WITH ME. That is MY son MINNNE. | hard non-toxic |
| The reason people become homebodies ... they are tired of being surrounded by idiots | hard non-toxic |
| Sick of seeing liberalism/conservatism is cancer. No, extremism on both sides is cancer and ur incompetence at realizing this is disgusting | hard non-toxic |
| I HAVE THE WORST AUTUMN EVER IN MY COUNTRY ITS FUCKING AUTUMN AND THIS IS FUCKING ANNOYING BECAUSE ITS KEEPS RAINING IN MY COUNTRY | hard non-toxic |
| buying pussy or smoking weed could get you more jail time than being a traitor to your countrymen apparently | hard non-toxic |
| Fellas we gotta invade Iran for feminism and gay rights. | hard non-toxic |
| What's Jesus's favorite gun? A nail gun. | hard non-toxic |
| Who is the only superhero that couId beat Captain America? Captain vietnam | hard non-toxic |
| I'm not weird, you're just a faggot | easy |
| Why is the monkey sad? Just learned that it will be a nigger in a million years. | easy |
| A red bull a day keeps the niggers away | easy |
| I honestly hate Indians, Muslims, really anything with brown, black or yellow coloured skin I hate. | easy |
| What was Adolf Hitler's biggest mistake? Not killing the 9,000,001 th Jew. | easy |
| Disclaimer: None of these things I'm about to tell you are 100% accurate. Just *mostly* accurate. There are always exceptions. | easy |
| countdown to more companies following suit after seeing the way that was handled. | easy |
| Thanks for the #RT! @marshawright @PrettyFru @cgm807 @poesravenlady @axelmojave @wallabemu7_62 Have a great week! | easy |
| that moment when you silence a mage and they start backpedalling and you just KNOW it's about to go down | easy |
| I still have one or two hunting knives from when I used to fish and shoot in case of self defence at home. | easy |

Table 3: 30 posts used in the online user study.

