# OpenReview forum: "BiasX: “Thinking Slow” in Toxic Content Moderation with Explanations of Implied Social Biases"
_EMNLP/2023/Conference — EMNLP 2023 Main_

### Official Review · Reviewer_rNFj · 2023-07-26

**Soundness:** 3

**Excitement:**

4: Strong: This paper deepens the understanding of some phenomenon or lowers the barriers to an existing research direction.

**Paper Topic And Main Contributions:**

This paper focuses on the problem of social biases, which indeed requires much attention in content moderation.
It introduces a framework that can help to improve content moderation quality.
The authors conduct a user study by crowdsourcing answers from more than 450 Mturk workers.
The main contributions are made by answering three questions: (1) this framework helps, (2) the explanations enable moderators to think more carefully, and (3) the quality of explanation matters.





**Questions For The Authors:**

A. What's the distribution (mean, variance for example) of labeling time in the qualification stage for each condition of workers?

B. Is there a pre-defined standard for crowd workers that they should follow when playing the role of moderator?

**Reasons To Accept:**

1. The framing of this problem is good and this paper is well-structured.

2. The user study is large with most parts well-designed.

3. It addresses the importance of adding explanations to difficult content for moderators.

**Reasons To Reject:**

1. My main reason to reject is the design to answer the 2nd research question: "Do biasx enable moderators to think carefully about moderation decisions".

First, the MTurk workers are randomly assigned to one of the four conditions. There may be chances that some workers performing tasks (even other tasks) very fast may fall in the same condition. Do the authors considering calculate labeling time for each worker during qualification stage, and re-calculate that in each condition to reduce individual labeling speed biases?

Second, while the intuitive makes sense, "think carefully" does not necessarily correlate with "labeling time". For example, some crowd workers may perform multiple tasks, switch between tabs, which may result in an increase in labeling time too. It would be better to base this on a theory, data or literature. Or mention this in the limitations.

2. "crowdworker are asked to play the role of content moderators", it should be better justified. The role, standard, or platform for moderators may vary a lot.

**Reproducibility:**

4: Could mostly reproduce the results, but there may be some variation because of sample variance or minor variations in their interpretation of the protocol or method.

**Reviewer Confidence:**

4: Quite sure. I tried to check the important points carefully. It's unlikely, though conceivable, that I missed something that should affect my ratings.

**Typos Grammar Style And Presentation Improvements:**

This paper overall is well written.

For figure 2: numbers in percentage are hard to see and can be a little larger.

For figure 4: text can be larger.

---

> ### Author Rebuttal · Authors · 2023-08-28
>
> We are glad that you like the framing of this work, and consider our study well-designed.
>
> **Outliers/Individual labeling speed differences could lead to inaccurate labeling time estimates**
>
> Like the reviewer pointed out, there are factors that could bias labeling time (e.g., individual speed differences, multi-tasking workers switching between tabs). This is precisely why we use the median labeling time statistic for each individual annotator, so that the measurement is more robust to outliers (e.g., if the worker took a break during one of the questions). We would like to further point out that we use a large sample size for our study (>100 participants in each group), and it is very unlikely that one group ends up with a substantially larger percentage of fast/slow annotators than another group. In addition, the 95% confidence intervals of the median labeling time measurements (Figure 2) are sufficiently tight for our results to be statistically valid even when individual speed differences are taken into account.
>
> **Careful thinking does not necessarily correlate with longer labeling time**
>
> Thank you for this acute observation. We are indeed using labeling time as one of the possible indicators for careful thinking, and we acknowledge that increased labeling time alone does not definitively suggest more careful reasoning. That said, in a controlled environment it’s reasonable to attribute this observed increase in time to more careful reasoning; the large sample size and computed statistics (e.g., tight 95% confidence interval) make it unlikely that our observations are due to noise. We want to further point out that the reported careful reasoning is further evident from the improvements in worker performance on hard instances when explanations are provided. We will elaborate on this discussion in our paper.
>
> **Standard for annotation**
>
> We do provide guidelines of annotation (e.g, label definitions and training examples) to all annotators, which can be found in Fig. 6, 7, 8 in the appendix. This standard for annotation roughly follows Reddit’s moderation guidelines. As we discussed in the limitations section (L335-350), we opt for a prescriptivist perspective [(Rottger et al., 2022)](https://aclanthology.org/2022.naacl-main.13/), which is often adopted by actual platforms’ terms of service (TOS) and community guidelines. Please see the first point in our response to Reviewer 7QKF for a related discussion.

---

### Official Review · Reviewer_5KD6 · 2023-08-04

**Soundness:** 5

**Excitement:**

3: Ambivalent: It has merits (e.g., it reports state-of-the-art results, the idea is nice), but there are key weaknesses (e.g., it describes incremental work), and it can significantly benefit from another round of revision. However, I won't object to accepting it if my co-reviewers champion it.

**Paper Topic And Main Contributions:**

The paper introduces BIASX, a novel framework aimed at assisting content moderators by providing free-text explanations of implied social biases in statements. Through a large-scale crowdsourced user study (N=454), it demonstrates that these explanations assist participants in correctly identifying subtly toxic content. However, the introduction of explanations leads to an increase in labeling time and mental demand, which could be potential drawbacks in real-world applications. The study also emphasize the superiority of human-written explanations over machine-generated ones. Finally, the authors highlight the challenges in identifying high-quality examples and generating human explanations, suggesting the need for further research.

**Questions For The Authors:**

Question A: How do you plan to address the challenge of identifying high-quality examples and generating human explanations, as mentioned in the limitation section?

Question B: What are the suggestions to improve the accuracy of the generated explanations in future iterations of the BIASX framework?

**Reasons To Accept:**

1. The paper demonstrates through a large-scale crowdsourced user study that participants benefit from these explanations in correctly identifying subtle toxicity. This empirical evidence strengthens the validity of the proposed framework.
2. The proposed framework - BIASX is very easy to implement enhances content moderation by providing free-text explanations of statements' implied social biases. This is a significant contribution to the field of content moderation and online safety.

**Reasons To Reject:**

A potential reason to reject this paper could be the relatively low accuracy of the generated explanations. The study demonstrates that only 60% of model explanations for hard toxic content are accurate, which could lead to downstream errors when applied in real-world content moderation scenarios. This imperfection in the model's performance might limit its practical utility and effectiveness.

**Reproducibility:**

4: Could mostly reproduce the results, but there may be some variation because of sample variance or minor variations in their interpretation of the protocol or method.

**Reviewer Confidence:**

4: Quite sure. I tried to check the important points carefully. It's unlikely, though conceivable, that I missed something that should affect my ratings.

---

> ### Author Rebuttal · Authors · 2023-08-28
>
> Thank you for your review, and we are delighted that you consider our framework easy to implement and empirically effective. We are really grateful that you consider this work a significant contribution to content moderation and online safety.
>
> **Low accuracy of the generated explanations and directions for improving accuracy**
>
> Indeed, explanations generated with GPT-3.5 have somewhat low accuracy on the hard-toxic instances (60%), which points to the need for combining our framework with models that are more capable of understanding toxicity. For example, GPT-4 reports 80% accuracy on the hard toxic instances, and thus we expect our framework to be even more helpful to moderators with GPT-4 explanations. Future work could also combine effective prompt strategies (e.g., chain-of-thought) with explanation generation to improve explanation quality.
>
> **Challenge of identifying high-quality examples**
>
> Based on [Han and Tsvetkov, 2020](https://aclanthology.org/2020.emnlp-main.622/), we use misclassification as a signal for example difficulty. While this approach is somewhat effective in filtering out the easy instances, the remaining instances contain a mix of mislabeled and ambiguous examples, which requires additional human verification. One practical way of identifying high-quality instances could be through dataset cartography [(Swayamdipta et al., 2020)](https://arxiv.org/abs/2009.10795), which characterizes individual instances using training dynamics.

---

### Official Review · Reviewer_7QKF · 2023-08-05

**Soundness:** 4

**Excitement:**

4: Strong: This paper deepens the understanding of some phenomenon or lowers the barriers to an existing research direction.

**Paper Topic And Main Contributions:**

The authors propose a framework for AI-generated explanations to assist users in content moderation. The idea is to support moderators with their decision-making by providing information about he entities and events of a given post. They authors conduct a user study and show that expert-written explanations support moderators while AI-generated explanations sometimes hinder correct moderation, i.e. explanation quality matters and AI models should be improved w.r.t. this.

**Questions For The Authors:**

- l. 055: What is a „hard … instance“ of bias? Please add more examples in the text in the beginning for all kinds of bias as this is a relevant point to understand not only on page 3 but from page 1.
- Could you add a more detailed explanation of what is understood as a „mental shortcuts“ right in the explanation section? In my opinion, it is not fully clear what exactly is meant by this term, how exactly this is grounded in scientific theory, etc.
- Could you please add a definition of subtle (e.g., 33, 67, 162, …) vs. overt bias (lexical overlap, etc.)? It is not really clear what the difference would be concretely in your work and how this difference affects data selection, the study, and the interpretation.
- It would be helpful to provide a little bit more details on what „dual process theory“ is, what the core notions are, and how these are leveraged / implemented.
- L. 171: I am not completely convinced of the process of using misclassified instances as „harder“ instances than correctly classified ones based on model performance only. How do you justify this? The model could be biased / bad at this for reasons other than toxicity is not easy to identify? How do you control for other phenomena which might be the real source of misclassifications?
- l. 177: Did you at least check results from one potential other LLMs (e.g. an open-source model)? Why did you use exactly this model only?
- Discussion section: Please elaborate a little bit more about human disagreement w.r.t accuracy — maybe there is no ground truth to agree on?
- Could please disentangle limitations, ethical considerations and the broader impact?

// edit: Thanks to the authors for their responses.

**Reasons To Accept:**

I like the approach and the idea and think that this work makes a focused and interesting contribution.
Based on the given information, the study  seems well-designed and carefully conducted. Another strength is the nuanced interpretation including side findings regarding the hard non-toxic examples.

**Reasons To Reject:**

I am a little concerned regarding bias in the generated explanations: Every human, and thus, also every model trained on human-generated text, contains bias. On a meta-level, specific perspectives of the world will still be amplified in explanations for content moderation, namely based on whatever bias a human expert or a trained model introduces into their explanation, e.g. values based on a paradigm / view of life where individualism is valued higher than the collective (which is not the case in every culture) or where democracy is the preferred political system (not true for all countries, maybe also not preferred by some cultures).
I would have liked to see at least a comment about the fact that these explanations still introduce a certain kind of bias (which could even be positive) which either needs to be accepted or addressed in further steps/future work.

Another point that I would to see addressed in a more nuanced way is defining what a subtle vs. toxicity is and how one can differentiate between both on which level (lexical, frame, ... levels). Here, more detailed definitions + references to related work should be added. In this regard, I am also missing a clearer interpretation of what a hard vs. an easy example of toxicity is and how the authors can be so sure that this holds for every use case. Here, I would highly recommend adding comments on limitations regarding whether a given post is toxic or not based on culture, subjective opinions, etc.

For scientific reproducibility, it would be very helpful if you shared. the code you used.

**Reproducibility:**

3: Could reproduce the results with some difficulty. The settings of parameters are underspecified or subjectively determined; the training/evaluation data are not widely available.

**Reviewer Confidence:**

3: Pretty sure, but there's a chance I missed something. Although I have a good feel for this area in general, I did not carefully check the paper's details, e.g., the math, experimental design, or novelty.

---

> ### Author Rebuttal · Authors · 2023-08-28
>
> Thank you for your thorough review and we are glad you find our work interesting and the study well-designed.
>
>
> **Generated explanations could be biased towards a certain view**
>
> We are glad that you pointed out the issue with biases in language models. This is, in part, what motivates our work on a human-AI collaborative moderation setup, rather than full delegation to a model. We acknowledge that there isn’t a universal set of guidelines that applies to different communities. That said, in practical moderation scenarios (e.g., Facebook, Twitter/X, or a specific subreddit), there are often prescribed or imposed viewpoints (e.g., by terms of service or moderation guidelines of the platform such as [Facebook's rules](https://web.archive.org/web/20230322180903/https://www.theguardian.com/technology/2018/apr/24/facebook-releases-content-moderation-guidelines-secret-rules)). As outlined by [Rottger et al., 2022](https://aclanthology.org/2022.naacl-main.13/), there are two contrasting study paradigms in toxicity annotations: a descriptive study, where variation in annotators’ views of biases is studied, and prescriptive study, with set guidelines that annotators follow; since most real world moderation settings are prescriptive, we opt for a prescriptive set of guidelines in our study as well. We will extend on this discussion in future versions of this work, and please see L335-350 (Limitations) for a relevant discussion.
>
> **How are “mental shortcuts by annotators” defined?**
>
> We follow definitions of *mental shortcuts* by [Tversky and Kahneman, 1974](https://pubmed.ncbi.nlm.nih.gov/17835457/): “mental shortcuts are heuristics that are employed in making judgements under uncertainty”. In particular, Tversky & Kahneman discuss the representativeness heuristic, which is very relevant to our work: when annotators are asked to label whether a post is toxic, they may use a heuristic of similarity to other toxic posts (e.g., swearwords causing annotators to flag toxicity, [(Sap et al., 2019)](https://aclanthology.org/2022.naacl-main.431/)). [Malaviya et al., 2022](https://aclanthology.org/2022.emnlp-main.438/) have further discussions connecting different heuristics to annotators for NLP tasks. We will add relevant context and definitions in the final paper.
>
> **How is subtle/hard vs. overt/easy toxicity defined?**
>
> We consider a post subtle/hard if it will likely be missed by an annotator using simple heuristics like the representativeness heuristic described above. For example, a *hard toxic* post may be sexist without derogatory slurs, and a *hard non-toxic* post may be expressing positive sentiment while containing profanity. We acknowledge that this classification could be fuzzy in real moderation settings, but the authors did our best to select unambiguous examples for our study. In addition, we would like to point out that the easy vs. hard classification is only intended for testing the effectiveness of our framework on different types of examples, and it has no bearing on the deployment of our system. We will clarify the classification of toxicity and incorporate this discussion into our paper.
>
> **What is dual process theory?**
>
> Dual process theory describes people’s thought and decision making arises from two processes: one “fast” and automatic, the other “slow” and deliberate. In the context of this work, “fast” refers to moderators using associative mental shortcuts for labeling. In contrast, “slow” refers to thoughtful moderation. That said, there isn’t a direct operationalization of the theory in our paper, and we don’t claim that hard toxic or hard non-toxic instances can only be correctly labeled using slow thinking. We will expand on relevant discussions in the paper.
>
> **Misclassified instances are not necessarily harder**
>
> Indeed, instances could be misclassified due to a variety of reasons other than instance difficulty (e.g., mislabeling, model biases). Due to exactly this, two authors went over all misclassified examples by the classifier and selected examples we agreed were hard, and can be unambiguously labeled (see L169-175). Thus, misclassifications caused by model biases do not influence the examples selected for our user study.
>
> **No ground truth to agree on**
>
> As mentioned previously, our work follows a prescriptive paradigm, under which we assume one correct label. During data selection, two authors verified and agreed that every instance can be unambiguously labeled under the moderation guidelines we provide to annotators (Figure 6 in the appendix contains the guidelines).
>
> **Results from one potential other LLMs**
>
> In preliminary experiments, we tried generating explanations with a fine-tuned T5-Large model, and found it substantially worse at spelling out the stereotypes in posts. Therefore, we chose GPT-3.5 (text-davinci-003), which was one of the best available LLMs at the time.
>
> **Code release and reproducibility**
>
> We will release the code for generating explanations with GPT-3.5 and toxicity classifier training in the final paper.

---

### Meta-Review · Area_Chair_fkUY · 2023-09-13

**Recommendation:** 5

**Metareview:**

This paper introduces a framework for AI-generated explanations to help improve content moderation quality, and they evaluate that framework with a large-scale crowdsourced user study to demonstrate that the framework helps moderators think more carefully during content moderation decisions.

Reviewers all agreed that this was a well-designed study and a well-written paper that makes a focused and interesting contribution to the NLP community and to work on content moderation and online safety, through the framework and the empirical results.

Reviewers wanted to see clearer explanations of the choice of the moderation framework (i.e., using a prescriptive framework, rather than a more descriptive approach to identifying bias) and the background of the participants in the study (i.e., that they were not themselves moderators). In addition, more discussion of the Tversky and Kahneman framework and more contextualization of what makes for difficult items to annotate would be useful for readers.

---

### Decision · Program_Chairs · 2023-10-07

**Decision:**

Accept-Main

**Comment:**

This paper introduces a framework for AI-generated explanations to help improve content moderation quality, and they evaluate that framework with a large-scale crowdsourced user study to demonstrate that the framework helps moderators think more carefully during content moderation decisions.

Reviewers all agreed that this was a well-designed study and a well-written paper that makes a focused and interesting contribution to the NLP community and to work on content moderation and online safety, through the framework and the empirical results.

Reviewers wanted to see clearer explanations of the choice of the moderation framework (i.e., using a prescriptive framework, rather than a more descriptive approach to identifying bias) and the background of the participants in the study (i.e., that they were not themselves moderators). In addition, more discussion of the Tversky and Kahneman framework and more contextualization of what makes for difficult items to annotate would be useful for readers.